# Reviewing and Discussing Graph Reduction in Edge Computing Context

**Asier Garmendia-Orbegozo [1,\*], José David Núñez-Gonzalez [2] and Miguel Ángel Antón [2]**

1 Department of Applied Mathematics, University of the Basque Country (UPV/EHU) Eibar, 207600 Eibar, Spain
2 TECNALIA, Basque Research and Technology Alliance (BRTA), 20001 San Sebastian, Spain
\* Correspondence: agarmendiao@ehu.eus

**Abstract:** Much effort has been devoted to transferring efficiently different machine-learning algorithms, and especially deep neural networks, to edge devices in order to fulfill, among others, real-time, storage and energy-consumption issues. The limited resources of edge devices and the necessity for energy saving to lengthen the durability of their batteries, has encouraged an interesting trend in reducing neural networks and graphs, while keeping their predictability almost untouched. In this work, an alternative to the latest techniques for finding these reductions in networks size is proposed, seeking to figure out a simplistic way to shrink networks while maintaining, as far as possible, their predictability testing on well-known datasets.

**Keywords:** graph reduction; edge computing; artificial intelligence; pruning

## 1. Introduction

The use of Deep Neural Networks (DNN) in different scenery such as image classification, voice synthesis or object detection is undoubtedly one of the most effective ways to make predictions. The development of DNNs in recent years has evolved in such a way that, nowadays, neural network designs have billions of parameters, with great capability of prediction, thus needing significant computation resources. Existing edge devices cannot satisfy tarting current demands, ranging from huge amounts of data that need to be stored safely to powerful computation units. However, by reducing the size of these architectures in an efficient way, their deployment in embedded systems could be feasible.

Among others, the most used and effective way to shrink these networks is through the use of techniques such as pruning and quantization. The former consists of removing parameters (neurons or weights) that have negligible contribution, while maintaining the accuracy of the classifier. On the other hand, quantization involves replacing datatypes with reduced-width datatypes, by transforming data to fit in new datatypes' shapes. In this way, reduced networks are able to compete with the original ones in terms of accuracy, even improving it in some cases in which overfitting issues were hindering their predictability. Moreover, by reducing the width of data the edge devices could face, the storage issue mentioned above could be solved and larger datasets in constrained memory sizes could be collected.

In this work a simplistic, heuristic attempt to achieve the reduction in size of these architectures is presented and it is compared with an exhaustive search for the same reduction in size of the former network. The results show that, with a considerable reduction in computation time, the pruned networks almost achieved the same performance metrics overall, after applying the two approaches. As there exists a lack of methods that offer a heuristic approach in a simplistic way to solving pruning tasks in these type of architectures, we aim to fill this gap in the literature. With a considerable reduction in computation time and an almost negligible loss of performance metrics, we aim to achieve the same reduction in network size using our approach. The rest of this paper is organized

as follows: Section 1 introduces and analyzes the literature of the pruning process and the most significant and attractive approaches made so far, similarly presenting some quantization techniques; Section 2 presents the simplistic approach cited above; in Section 3, experiments are described and their results are outlined; and, finally, Section 4 concludes this work and proposes some possible future research lines.

### 1.1. Literature Review
Pruning

Pruning consists of removing part of the connections (weights) or neurons from the original network so as to reduce the dimensions of the original structure while maintaining its ability to predict as shown in Figure 1. The core of this technique resides in the redundancy that some elements add to the entire architecture. Memory size and bandwidth reduction are addressed with this technique. Redundancy is lowered and overfitting is faced in some scenarios. Different classifications of works based on this ability are made depending on:

- Element pruned;
- Structured/Unstructured;
- Static/Dynamic,

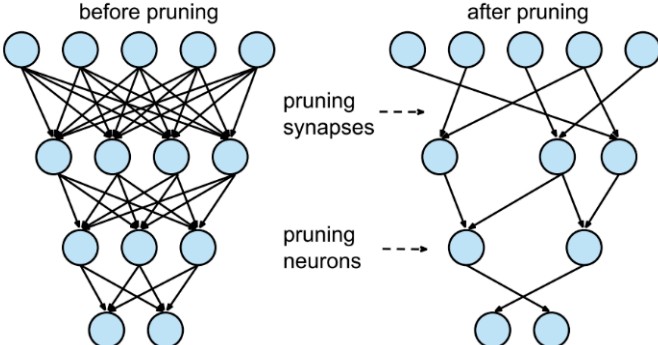

**Figure 1.** Different approaches to pruning. Source: https://towardsdatascience.com/pruning-deep-neural-network-56cae1ec5505 (accessed on 20 July 2022).

The element pruned can be either a connection or a neuron in a pruning process. The difference between structured and unstructured pruning lies in whether the pruned network is symmetric or not. Filter-wise pruning, which removes all parameters belonging to one or more filters, is a good example of a structured pruning approach. When we talk about static pruning, we refer to the process in which all pruning steps are made before the inference time, whereas dynamic pruning is performed during runtime.

### 1.2. Static Pruning

Static pruning is the process of removing elements of a network structure offline, before training and inference processes. During these last two processes, no changes are made to the network previously modified. However, after removing different elements of the architecture, a fine-tuning or retraining of the pruned network is interesting. This is due to the changes that suffer the network by removing big part of its elements. Thus, some computational effort is needed in order to reach comparable accuracy to the original network.

The pruning has been carried out by following different criteria: Some works were based on the magnitude of the elements themselves. It is undoubtedly true that near-zero values of weight make far less contribution to the results than others that surpass a certain threshold value. Thus, by removing connections that may appear unneeded, the original network is shrunk. It is interesting to develop this process layer-by-layer while not affecting the performance of the resulting network. By removing elements from the entire network,

some connections or neurons may take on a different role in the resulting network, thus some fine-tuning or retraining may occur.

In [1,2], they used the second derivative of the Hessian matrix to reduce the dimension of the original architecture. Optimal Brain Damage (OBD) and Optimal Brain Surgeon (OBS), respectively, work under three assumptions. Quadratic: the cost function is near quadratic. Extremal: the pruning is done after the network converged. Diagonal: sums up the error of individual weights by pruning the result of the error caused by their co-consequence. Additionally, OBS avoids the diagonal assumption and improves neuron removal precision by an up-to-90% reduction in weights for XOR networks. Taylor expansions of first order were also considered to reduce the network dimension in [3,4], as a criterion to approximate the loss in the objective function as an effect of pruning.

Some other works have followed the idea of removing elements based on different penalization terms. Penalty-based training aims to modify or add an error function to modify weights during training process using a penalty value. At the end, near-zero values are pruned from the original network. LASSO [5] was introduced as a penalty term it shrinks the least-absolute-valued feature's corresponding weights, increasing weight sparsity. This operation has been shown to offer a better performance than traditional procedures, such as OLS, by selecting the most significantly contributed variables instead of using all the variables, achieving approximately 60% more sparsity than OLS. The problem with LASSO is that it is an element-wise pruning technique, leading to unstructured networks and sparse weight matrices. By performing this technique group-wise, as Group LASSO does [6], removing entire groups of neurons and maintaining the original network's structure, this last issue was solved. Groups are made based on geometry, computational complexity or group sparsity, among others.

Other alternatives have been proposed for carrying out static pruning. In [7] a novel criterion for Convolutional Neural Network (CNN) pruning was proposed, called layer-wise relevance propagation. It measures the contribution of each unit to the relevance of the decision making. In this way, the units that are below a predefined threshold are removed from the graph and, consequently, the relevance of each unit is recomputed. For this last step, the total relevance per layer is calculated in order to keep it untouched during iterations. Thus, each unit's relevance is recalculated to maintain this value.

In [8], a technique for pruning redundant features, along with their related feature maps, according to their relative cosine distances in the feature space was proposed; thus, leading to smaller networks with reduced post-training inference computational costs and competitive performance. Redundancy can be reduced while inference cost (FLOPs) is reduced by 40% for VGG-16, 28–39% for ResNet-56/110 models trained on CIFAR-10, and 28% for ResNet-34 trained on ImageNet database, with minor loss of accuracy. To recover the accuracy after pruning, models were finetuned for a few iterations without the need to modify hyper-parameters.

In [9], combining the ideas of sparsity and the existence of unequal contributions of neurons towards achieving the target, the sparse low rank (SLR) method is presented; this method sparsifies Single Value Decomposition (SVD) matrices to achieve a better compression rate by keeping lower ranks for unimportant neurons. In this way, it is possible to save $3.6\times$ the storage space of SVD without much effect on model performance. The structured sparsity achieved by the proposed approach also has the advantage of speeding up the computation.

Another interesting approach to be taken into consideration is pruning filter-by-filter. Filter-wise pruning [10] uses the l1-norm to remove filters that do not affect the accuracy of the classification. Pruning entire filters and their related feature maps resulted in a reduced inference cost of 34% for the VGG-16 model and 38% for the ResNet-110 model on the CIFAR-10 dataset, with improved accuracy of 0.75% and 0.02%, respectively. ThiNet [11] adopts statistical information from the next layer to determine the importance of each filter. It uses a greedy search to prune the channel that has the smallest reconstruction cost in the next layer. During each training, pruning is carried out more lightly to allow for coefficient

stability. The pruning ratio is a predefined hyper-parameter, and the runtime complexity is directly related to the pruning ratio. ThiNet compressed ResNet-50 FLOPs to 44.17% with a top-1 accuracy reduction of 1.87%.

Other research has been carried out on activations, which may also be indicators for pruning corresponding weights. Average Percentage of Zeros (APoZ) [12] was introduced to judge if one output activation map was contributing to the result. Some activation functions, particularly rectification functions such as Rectified Linear Unit (ReLU), may result in a high percentage of zeros in activations being relevant to their pruning.

After applying different techniques to reduce the amount of non-relevant elements from the original structure, it is essential that there be a fine-tuning or retraining phase. It has been shown [8] that by training a pruned structure from scratch, less accurate results are obtained compared to the retraining processes in which weights from the original network are maintained for the new training phase. That is why, iteratively, a retraining or fine-tuning step follows after each pruning step has been carried out. This iterative process is repeated until a desired number of elements has been achieved.

*1.3. Dynamic Pruning*

Pruning a DNN dynamically offers several benefits compared to the same process carried out offline before both training and inference processes. Identifying at runtime which elements of the original structure are relevant and which ones are not, offers the possibility of solving different issues related to static pruning by adapting the network to the changes of input data. However, this process is far more complex than the static one, so various decisions need to be made before starting it. In some cases, it makes sense to consider additional networks or connections to further assist the pruning process. Information input could be either layer-by-layer, feeding a window of data iteratively to the decision system, or by one-shot feeding. Like in static pruning, a score system and a comparative system (automatic or manual) must be established. Similarly, a stopping criterion must be imposed, and finally, the additional components have to be trained at the same time as the network is trained.

A negative impact on the system computation requirements also needs to be taken into account. Additional bandwidth, computation and power resources are necessary while computing at runtime which elements are to be pruned. At the same time, convolutional operations with large number of features consume a huge part of the bandwidth. Thus, a trade-off between dynamic pruning overhead, reduced network computation, and accuracy loss, should be considered. Different approaches have been developed in recent years, and the most significant ones are described below.

In [13,14], they focused on conditional computing by activating relevant parts of the original network. The non-activated elements act as pruned ones, enlightening the original structure.

In [15–17], different alternatives of cascade network were proposed. A cascade network consists of a series of networks that each of which has its own output layer, instead of offering an output per layer. Its main advantage is that it can offer an early exit if desired accuracy is achieved. In contrast, some hyper-parameters need to be tuned manually. Moreover, in [18] Blockdrop was introduced as a reinforcement learning method that, with an input image, was able to deduce which blocks should participate in the whole process. They were able to achieve an average speed-up of 20% on ResNet-101 for ILSVRC-2012 without accuracy loss. On the other hand, Runtime Neural Pruning (RNP) has been proposed [19], based on a feature selection problem, as a Markov Decision Problem (MDP) when finding computation efficiency. A Recursive Neural Network (RNN) based network was used to predict which feature maps were necessary. They found $2.3\times$ to $5.9\times$ reduction in execution time with top-5 accuracy loss from 2.32% to 4.89% for VGG-16.

In [20], a novel dynamic pruning technique, based on pruning and splicing, was presented. On one hand, pruning operations can be performed whenever the existing connections seem to become unimportant. On the other hand, the mistakenly pruned

connections will be re-established if they once appear to be important (splicing). Experimental results show that their method compressed the number of parameters in LeNet-5 and AlexNet by a factor of $108\times$ and $17.7\times$, respectively, with a better learning efficiency.

The negative point of RL techniques is their computation expense. Alternatively, differentiable approaches have been made to solve this issue. Using Dynamic Channel Pruning (DCP), in [21] they proposed a side network called Feature Boosting and Suppression (FBS) to decide which channel to skip. FBS achieved $5\times$ acceleration on VGG-16 with 0.59% ILSVRC-2012 top-5 accuracy loss, and $2\times$ acceleration on ResNet-18 with 2.54% top-1, 1.46% top-5 accuracy loss. Similarly, in [22], a channel-threshold weighting decision (T-Weighting) was used to prune channels dynamically. A T-sigmoid activation function, using as its entry a downsampling from a Fully Connected Layer (FCL), was used to calculate channels' scores and decide which ones to prune.

Another interesting approach to dynamic pruning of CNNs has been proposed in [23]. They explored the manifold information in the sample space to discover the relationship between different instances from two perspectives, i.e., complexity and similarity, and whether the relationship was preserved in the corresponding sub-networks. An adaptive penalty weight for network sparsity was developed to align the instance complexity and network complexity, while the similarity relationship was preserved by matching the similarity matrices. Extensive experiments were conducted on several benchmarks to verify the effectiveness of this method. Compared with the state-of-the-art methods, the pruned networks obtained by this method can achieve better performance with less computational cost. For example, they can reduce 55.3% FLOPs of ResNet-34 with only 0.57% top-1 accuracy drop in ImageNet.

In [24], a multi-objective-based pruning technique for CNNs called CURATING was presented. This method retains filters that are less redundant in terms of their representation, have a high saliency score and are likely to produce high activations. On a range of CNNs over well-known datasets, CURATING exhibits a better or comparable tradeoff between model size, accuracy, and inference latency than state-of-the-art methods. For instance, pruning VGG16 on the ILSVRC-12 dataset, CURATING achieves higher accuracy and a smaller model size than previous techniques.

Alternatively, the latency-aware automatic CNN channel pruning method (LACP) was introduced in [25], seeking to find low latency and accurate pruned network structure automatically. They evaluated the inaccuracy of measuring pruning quality by FLOPs, and a number of other parameters and used model inference latency as the direct optimization metric. To bridge model pruning and inference acceleration, they analyzed the inference latency of convolutional layers on GPU. Results showed that the inference latency of convolutional layers exhibits a staircase pattern in relation to channel number due to the GPU tail effect. Experiments and comparisons with state-of-the-art methods on three image classification datasets show that this method can achieve better inference acceleration with less accuracy loss.

### 1.4. Quantization

Reducing the number of bytes used to represent data is crucial when transferring ML algorithms to resource constrained edge devices. When we mention quantization, we refer to the transformation of data from floating-point representation to a determined value range. These former data could be represented either by predefined values or symbols. In the context of DNNs, weights may be quantized by clustering processes, by lookup tables or by using linear functions, all of these with the aim of reducing the information width of data. Originally, most of the DNNs applied 32-bit floating-point representation for weight parameters but, in most cases, this is excessive. In fact, 8-bit values can significantly accelerate inference processes without observable loss in accuracy. Different alternatives have been adopted in recent years, and some of the most interesting ones are described and analyzed below.

In [26], they conducted extensive experiments using incremental quantization on three applications: medical image segmentation, image classification, and automatic speech recognition. The main goal of incremental quantization was to convert 32-bit-floating-point weights (*W*) into low-precision weights *(W')*, either power of two or zero with minimum accuracy loss. Each *W'* is chosen from $Pl = \{\pm 2^{n1} \ldots \pm 2^{n2}\}$, here *n*1 and *n*2 are two integer numbers determined by the max absolute value of *W* of each layer and expected quantized bit-width, and $n2 \leq n1$. Experimental results in FCN for biomedical image segmentation, CNN for image classification, and RNN for automatic speech recognition, show that incremental quantization can improve accuracy by 1%, 1.95%, and 4.23% on the three applications, respectively, with $3.5\times$ to $6.4\times$ memory reduction.

An adaptive quantization method was introduced [27] to enhance quantization results on MNIST, CIFAR-10 and SVHN datasets, finding a unique, optimal precision for each network parameter such that the increase in loss was minimized. Pruning unnecessary parameters or quantizing them showed that DNN model sizes can be reduced significantly without loss of accuracy. The resulting models were significantly smaller than in state-of-the-art quantization techniques.

Following the trend of mixing pruning and quantization techniques, in [28], they presented a training acceleration framework able to speed up the training process while compressing DNN for mitigating overhead transmission. FL-PSQU is a Federated Learning mechanism that is divided in three steps. First, a one-shot pruning is done by the server to generate general models for all clients and, after quantizing it, it is transferred to each client. Then, each client updates its model and depending on its update's significance, it is transmitted to the server or not; in this way unnecessary communications are avoided.

Other simpler approaches have been made that include binary and ternary quantization. Thus, only two or three possible values were assigned to each element of the architecture with a vast reduction in memory size and computation effort. However, accuracy loss is not negligible in these techniques due to the hard generalization of weights. Binary Connect [29] used stochastic plus–minus quantization by assigning +1 to positive-valued weights and −1 to negative-valued ones with hard sigmoid probability σ(x) and 1 − σ(x), respectively. Similarly, in [30] they binarized all weights of different architectures and afterward multiplied with a scaling factor for all the weights in a layer. The authors of [31,32] showed the power of ternary quantization by including additional values (−1, 0 and +1) compared to the binary case. Its implementation in hardware must be efficient due to the fact that the 0 value does not actually participate in computations. Ternary Weight Networks (TWN) adopt the l2-distance to find the scale and formats the weights into −1, 0 and +1, with a threshold generated by an assumption that the weighs are uniformly distributed, such as in [−a, a]. This resulted in up to $16\times$ model compression with 3.6% ResNet-18 top-1 accuracy loss on ILSVRC-2012.

Some scenarios could be of special interest the conversion of floating-point multiplication to bit-shift operations, such as in the case of FPGAs. Here, the constraints for weights being the power of two, leverages training and inference effort and time. This approach was proposed in [33] by quantizing the representation of weights layer-by-layer. Likewise, Incremental Network Quantization [34] replaces all weights with powers of two, iteratively, in each iteration, preserving some weights in full precision and retraining them. After multiple iterations, the majority of weights are converted to power of two. The final structure has weights from two to five bits, with values near zero set to zero. The results of group-wise iterative quantization show lower error rates than a random power-of-two strategy.

## 2. Materials and Methods

### 2.1. Heuristic Partial Search

Our approach is mainly focused on reducing the size of the original network by pruning some of the connections between layers. We had focused on removing part of the connections of the original architecture based on the score obtained after removing each

part of it, i.e., comparing the accuracies of the original network and the modified network. The weights that had the best score, i.e., the ones that had the most pronounced decay in the accuracy of the network after removing them, are preserved, since they are the most relevant ones. However, finding the subset of weights that best preserved the accuracy of the original network is an exhaustive process that carries huge computational effort and time. In contrast, with the Heuristic Partial Search (HPS) proposed here this search is far more feasible than the previous one in terms of computational costs.

When we aim to find the subset of the most relevant weights to preserve in the original network structure, instead of calculating, one by one, each weight's score we proposed finding the most relevant weights in the following way. Given that a Fully Connected (FC) layer of m inputs and n outputs is represented by an $m \times n$ dimension matrix, we find row by row the subset of weights that best preserves the original networks' accuracy tested on a well-known dataset. First, we find the subset of weights that have the best score in an exhaustive way. Then, we try to find the subset of most relevant weights in the following way:

- Compare the score of the 1st weight and $(n/2 + 1)$th weight.
- If the 1st weight is less relevant compare it with the $(n/4 + 1)$th weight. Otherwise, compare the $(n/2 + 1)$th weight with the $(3n/4 + 1)$th weight.
- Find, in a predefined number of iterations, the weight that has the best score for each row and remove it from the original network.
- Complete the process until the desired reduction in the network is achieved for each row, until the predefined number of eliminations is achieved. Successively repeat the process for the remaining rows until the entire layer has been pruned.

The pseudo-code of the algorithm is given in Algorithm 1.

Alternatively, we carried out an exhaustive search for the most relevant weights, as mentioned above. Instead of finding in a predefined number of iterations the best subset of weights for each row of the weights' matrix, we tested each weight's score and, in this way, obtained the best subset of weights. In the next section the architecture and datasets that were utilized are described in depth; details about the election of hyper-parameters are also given.

---

**Algorithm 1.** Pseudo-code of the Heuristic Partial Search (HPS).

---

```
For a in nrows:
    For b in neliminations:
        top_score = 0
        step = ncolumns//2
        For c in niterations:
            update pos1 and pos2
            weights_copy[pos1] = 0
            set model weights
            evaluate the model => score1
            weights_copy2[pos2] = 0
            set model weights
            evaluate the model => score2
            if score2 < score1:
                if top_score < score1:
                    top_score = score1
                    best_pos = pos1
            else:
                if top_score < score2:
                    top_score = score2
                    best_pos = pos2
        original_weights[best_pos] = 0
    evaluate the pruned model
End
```

---

## 2.2. Experimental Settings

The architecture used to demonstrate the validity of the proposed approach in the case of CNNs was Lenet5, applied on the well-known dataset MNIST, a large database of handwritten digits that is commonly used for training various image processing systems. It contains 60,000 training images and 10,000 testing images of $32 \times 32$ pixel size. Alternatively, a network that consisted of 3 FC layers, each of them composed of 100 neurons, was tested on an IMDB dataset, which contains 25,000 movie reviews labeled by sentiment (positive/negative). In this way, our approach was tested and compared with an exhaustive search in image classification tasks, testing in a simpler scenario with a binary decision task as well. Network specifications are summarized in Tables 1 and 2.

**Table 1.** Specifications of Lenet5 architecture.

| Layer Name | Layer Type | Feature Map | Output size | Kernel Size | Stride | Activation |
|:---:|:---:|:---:|:---:|:---:|:---:|:---:|
| Input | Image | 1 | $32 \times 32 \times 1$ | - | - | - |
| Conv1 | 1xconv | 6 | $28 \times 28 \times 6$ | $5 \times 5$ | 1 | tanh |
| Pool1 | Avg pooling | 6 | $14 \times 14 \times 6$ | $2 \times 2$ | 2 | tanh |
| Conv2 | 1xconv | 16 | $10 \times 10 \times 16$ | $5 \times 5$ | 1 | tanh |
| Pool2 | Avg pooling | 16 | $5 \times 5 \times 16$ | $2 \times 2$ | 2 | tanh |
| Flatten | Flatten | - | 400 | - | - | tanh |
| FC3 | Dense | - | 120 | - | - | relu |
| FC4 | Dense | - | 84 | - | - | relu |
| FC5 | Dense | - | # classes | - | - | softmax |

**Table 2.** Specifications of second architecture.

| Layer Name | Layer Type | Feature Map | Output Size | Kernel Size | Stride | Activation |
|:---:|:---:|:---:|:---:|:---:|:---:|:---:|
| Input | Image | 1 | $10,000 \times 1$ | - | - | - |
| FC1 | Dense | - | $100 \times 1$ | - | - | relu |
| Dropout1 | Dropout | | | | | |
| FC2 | Dense | | $100 \times 1$ | - | - | relu |
| Dropout1 | Dropout | - | - | - | - | - |
| FC3 | Dense | | # classes | - | - | sigmoid |

The number of eliminations for each of the pruning processes was determined depending on the pruning rate desired. The number of iterations used to find the local optima by HPS was fixed at 6 iterations, given the sizes of the FC layers to be pruned.

## 3. Results

Figure 2 shows the accuracies obtained after pruning an FC layer in Lenet5 architecture tested on MNIST dataset. As far as could be observed, the difference between pruning, after an exhaustive search of the least significant set of weights to remove from the original network, and the iterative process was far less significant than expected. There was a negligible difference between both methods, at the expense of much longer computation time for different pruning rates, as shown in Table 3.

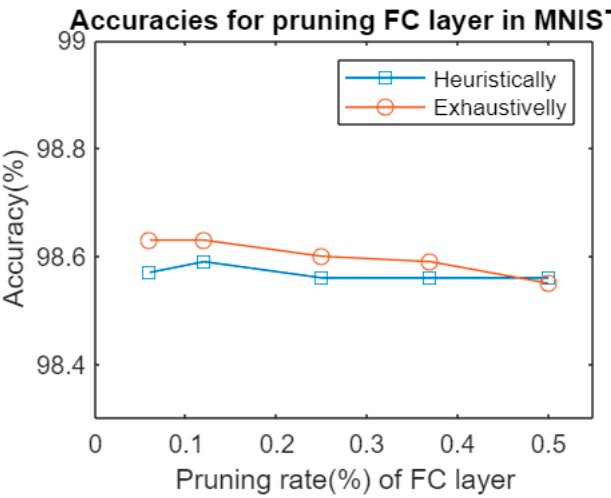

**Figure 2.** Accuracies for pruning an FC layer in Lenet5 HPS vs. pruning exhaustively.

**Table 3.** Computation time in seconds for the pruning process in Lenet5 architecture.

| Method | Pr = 0.06 | Pr = 0.125 | Pr = 0.25 | Pr = 0.37 | Pr = 0.5 |
|--------|-----------|------------|-----------|-----------|----------|
| Exh.   | 2.91      | 5.02       | 9.97      | 15.56     | 22.57    |
| HPS    | 1.18      | 1.33       | 1.74      | 2.19      | 2.77     |

There were similar findings in the network composed of three FC layers when testing on an IMDB dataset. The differences between both methods were not significant, but the computation effort differed completely between them, as summarized in Table 4. Figure 3 shows the accuracies for pruning using both alternatives.

**Table 4.** Computation time in seconds for the pruning process in the alternative architecture.

| Method | Pr = 0.05 | Pr = 0.1 | Pr = 0.2 | Pr = 0.35 | Pr = 0.5 |
|--------|-----------|----------|----------|-----------|----------|
| Exh.   | 14.84     | 29.28    | 55.93    | 97.74     | 142.19   |
| HPS    | 5.12      | 10.12    | 19.43    | 34.4      | 50.25    |

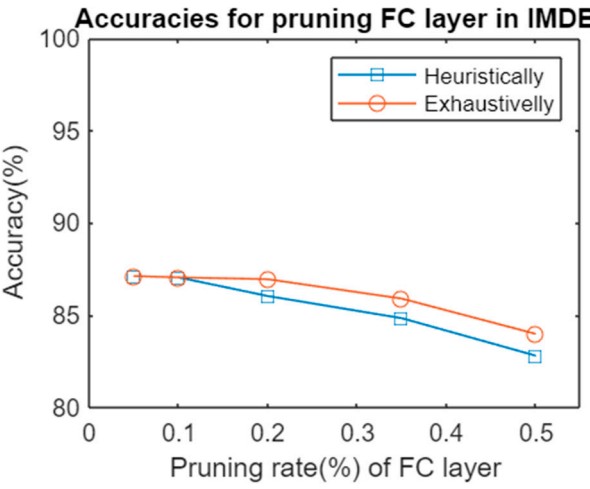

**Figure 3.** Accuracies for pruning an FC layer in the alternative architecture heuristically vs. pruning exhaustively.

## 4. Conclusions

When transferring deep networks usually trained on cloud to edge devices, it is of great interest to reduce the network size to adapt these architectures to the constraints of such devices. Thus, pruning the original network is undoubtedly an essential step to fit deep models in resource-constrained devices. Among different approaches to reduce the dimensions of the original network size and optimize inference time that have been made so far in the literature, adopting an exhaustive search of the most significative connections between different layers offers close- to-optimal results with low computational costs. As presented in this work, there is a negligible difference between pruning exhaustively and applying HPS. However, the time efficiency that HPS offers should be taken into consideration by end-users when transferring these models into edge devices due to the big differences shown in chapter IV. In some applications, where accuracy of these models is not critical, applying a heuristic approach to reduce the size of the original architecture makes senses, due to the high computational load that is needed to work out these reduction tasks. The results obtained by HPS do not outperform the ones obtained by an exhaustive search of the optimal set of connections between layers. The local optimal point found by our algorithm may not be as good as the global one offered by the general exhaustive search. However, the nature of these networks demands a non-excessive computational effort and high time efficiency, to avoid computations that might need weeks to complete.

In contrast, quantization techniques should not be underestimated, especially when transferring models to microcontrollers that lack operating systems. Particularly when those microcontrollers own FPGAs, conversion from floating-point multiplications to bit-shift operation could alleviate great computational cost and also reduce runtime. In the majority of scenarios, reducing the number of representative bits to eight is enough to avoid a significant loss in terms of accuracy while vastly reducing memory usage. Nonetheless, some other approaches such as binary quantization are too hard, in many cases leading to an excessive accuracy drop.

As a future possible work, we could look for a more optimal way to reduce different types of network heuristically, with no losses in accuracy, while maintaining the same rate of reduction in both computational effort and memory usage.

**Author Contributions:** All authors contributed equally. All authors have read and agreed to the published version of the manuscript.

**Funding:** This research received no external funding.

**Data Availability Statement:** Publicly available datasets at https://keras.io/api/datasets/ (accessed on 21 January 2022).

**Acknowledgments:** This work is partially supported by H2020-MSCA-RISE-2017 No 777720, Cyb-SPEED: Cyber-physical Systems for Pedagogical Rehabilitation in Special Education.

**Conflicts of Interest:** The authors declare no conflict of interest.

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
