# Peer review of "Reviewing and Discussing Graph Reduction in Edge Computing Context"

_computation, doi:10.3390/computation10090161_

Round 1

Reviewer 1 Report

The paper discusses the advantages of pruning a neural network to improve storage and performance on edge devices. While the topic is interesting, the actual contribution of the paper is limited. Most of the paper discusses existing techniques for pruning and quantization (the latter is not linked to the authors' contribution). The contribution of the work is limited to a heuristic method that is used to prune a network and shown to perform comparably to an exhaustive network on 2 randomly selected public datasets. The approach does not have any mathematical backing and hasn't been compared to the performance of exhaustive networks in the studies cited in related work. It is, therefore, inconclusive whether the heuristic approach can be generalized or not. The paper does a good job in the literature review section. The grammar can be improved in places. Overall the contribution to scientific knowledge is low to none. 

Author Response

Dear Reviewer,

This work’s contribution to the literature is given now in the introductory part, as it was absent previously. All tables are referenced except Table 4, so that change has been made. There were some swapped data, and that mistake has been corrected.  A brief explanation of the performance offered by our approach is given in Conclusions’ section. The pseudocode of our algorithm we think it is sufficient to understand our approach’s action.

Yours faithfully,

Best regards.

Asier Garmendia Orbegozo

Reviewer 2 Report

Dear Editor,

This paper aims to develop graph reduction in edge computing. But this paper lacks novelty in algorithmic steps and needs to apply recent comparative results. 

Author Response

(The authors gave the same response as above.)

Reviewer 3 Report

Dear Authors,

The manuscript of "Reviewing and discussing graph reduction in Edge Computing context" is very good. However, I have some review comments:

1)    all figures alignment is not centering.  

2)    The authors used MNIST dataset. But how many data size are there?

3)    The conclusion of this manuscript is needed to short with concise explanation.

4)    Also, the authors should mention the future research direction in conclusion section.

5)    The algorithm is written properly, the input parameters are missing there.

Author Response

Dear reviewer,

All figures are well aligned from our point of view. The data size of MNIST is given in the revised version. The conclusion section has been modified. The input parameters of the algorithm are dependent of the dataset and network of each case, so it is preferable to avoid including them in our opinion.

Best regards.

Asier Garmendia Orbegozo